# The Effects of Single or Combined Supplementation of Probiotics and Prebiotics on Growth Performance, Dietary Energetics, Carcass Traits, and Visceral Mass in Lambs Finished under Subtropical Climate Conditions

**DOI:** 10.3390/biology10111137

**Published:** 2021-11-05

**Authors:** Alfredo Estrada-Angulo, Octavio Zapata-Ramírez, Beatriz I. Castro-Pérez, Jesús D. Urías-Estrada, Soila Gaxiola-Camacho, Claudio Angulo-Montoya, Francisco G. Ríos-Rincón, Alberto Barreras, Richard A. Zinn, José B. Leyva-Morales, Xiomara Perea-Domínguez, Alejandro Plascencia

**Affiliations:** 1Faculty of Veterinary Medicine and Zootechnics, Autonomous University of Sinaloa, Culiacan 80260, Mexico; alfred_vet@uas.edu.mx (A.E.-A.); octaviozapata.fmvz@uas.edu.mx (O.Z.-R.); isabel.castro@uas.edu.mx (B.I.C.-P.); david.urias@uas.edu.mx (J.D.U.-E.); soilagaxiola@uas.edu.mx (S.G.-C.); c.angulom@uas.edu.mx (C.A.-M.); fgrios@uas.edu.mx (F.G.R.-R.); 2Veterinary Science Research Institute, Autonomous University of Baja California, Mexicali 21100, Mexico; abarreras@uabc.edu.mx; 3Animal Science Department, University of California, Davis, CA 95616, USA; razinn@ucdavis.edu; 4Health Sciences Department, University Autonomous of the West, Guasave 81048, Mexico; jose.leyva@uadeo.mx (J.B.L.-M.); xiomara.perea@uadeo.mx (X.P.-D.); 5Natural and Exact Sciences Department, University Autonomous of the West, Guasave 81048, Mexico

**Keywords:** probiotics, prebiotics, finishing lambs, performance, carcass, visceral mass

## Abstract

**Simple Summary:**

Concern about the use of antimicrobial compounds in livestock production has promoted research of “generally recognized as safe” additive alternatives. Probiotics (living microorganisms) and prebiotics (certain type of carbohydrates derived from yeast) have been shown to alleviate the negative effects of stress and boost immunity, thereby enhancing efficiency of energy utilization. In some regions (i.e., tropical and arid zones), livestock experience adverse climatic conditions, including elevated ambient temperature and humidity, which affect their productivity. Supplementation with probiotics and prebiotics may help to alleviate these adverse effects. In the present study, supplemental probiotics or/and prebiotics improved dietary energetic efficiency in lambs finished under subtropical climatic conditions. The combination of probiotics with prebiotics reinforced this positive effect.

**Abstract:**

The aim of this trial was to test the effects of the use of eubiotics (pro- and prebiotics) alone or in combination in the diet of lambs finished under subtropical climate conditions. For this purpose, 40 Pelibuey × Katahdin lambs (29.5 ± 4.8 kg initial live weight) were used in a 93 day growth-performance experiment. Dietary treatments consisted of a cracked corn-based finishing diet supplemented with (1) no eubiotics (control), (2) 3 g of probiotics (live *Saccharomyces cerevisiae*, SC), (3) 3 g of prebiotics (mannan oligosaccharide plus b-glucans, MOS), and (4) a combination of 1.5 g of SC and 1.5 g of MOS (SC+MOS). Throughout the study, the average temperature humidity index (THI) was 78.60. Compared to controls, supplementation with SC or MOS, alone did not affect average daily gain (ADG), but enhanced feed efficiency by 5.6% and 6.9% (gain-to-feed ratio, G:F) and dietary net energy by 4.6% and 5.9%, respectively. Compared to controls, SC+MOS enhanced ADG (10%), G:F (9.5%), and dietary net energy (7.2%). Lambs fed SC+MOS had also greater ADG, G:F, and dietary net energy compared to lambs fed SC alone. When compared to MOS, the combination enhanced ADG (10.4%, *p* = 0.04). This effect could be attributed to the increased dry matter intake (7.6%, *p* = 0.06), as neither G:F nor dietary energy was significantly affected. Compared with controls and SC, supplementation with MOS alone and SC+MOS increased kidney–pelvic–heart fat, while SC supplementation tended (*p* = 0.08) to reduce 4.1% the relative intestinal mass (as a proportion of empty body weight) when compared to controls. Treatment effects on the other carcass measures were not significant. In the present study, supplemental probiotics and/or prebiotics improved dietary energetic efficiency in lambs finished under subtropical climatic conditions. The combination of probiotics with prebiotics reinforced this positive effect.

## 1. Introduction

Among the strategies to reduce the negative effects of global warming on the productivity and health of livestock is antibiotic supplementation at a subtherapeutic level. In this sense, in cattle finished under high environmental heat load, the use of antibiotics such as ionophores has shown benefits related to a reduction in energy requirements [1].

However, the use of antibiotics as feed additives (growth promoters, AGPs, and ruminal fermentation manipulators, among others) is increasingly restricted around the world [2]. Probiotics and prebiotics are promising alternatives to conventional antibiotic supplementation [3,4]. Probiotics (live beneficial bacteria and/or yeast) and prebiotics (mainly fructo-oligosaccharides and galacto-oligosaccharides derived from indigestible carbohydrates from yeast cell wall), globally named eubiotics, have antimicrobial properties, inhibit proliferation of enteric pathogens, and may enhance intestinal health [5]. Probiotics and prebiotics act in a distinct manner in the gastrointestinal tract, which, under certain conditions, could be complementary. Accordingly, when the two are supplemented together, the combination is referred to as “synbiotics”. This combination of probiotics with prebiotics resulted in greater reductions in morbidity and mortality of dairy calves than when each was solely supplemented [6]. Under the same climatic conditions in which this experiment was conducted, finishing lambs daily supplemented with a combination of 1.5 g of live *Saccharomyces cerevisiae* plus 1.5 g of mannan oligosaccharide had greater total tract digestion of NDF and N, increased ruminal VFA concentration, and decreased ruminal concentration of hyper-ammonia ruminal bacteria than when fed with a dose of 3 g/lamb/day of each eubiotic [7]. Additionally, synbiotic supplementation may induce metabolic changes (increased plasmatic glucose, reduced cortisol levels, and reduced cellular oxidative stress) that promote efficiency of energy utilization under stress conditions [8,9,10]. These potentiating effects may be of particular benefit to lambs reared in subtropical and tropical environments. However, very limited information is available regarding effects of the combination of probiotics with prebiotics on growth performance, dietary energetics, and carcass characteristics of fattening lambs under conditions of high ambient temperature and humidity. Accordingly, the objective of this experiment was to evaluate the effects of single or combined dietary supplementation of probiotics and prebiotics on growth performance, dietary energetics, and carcass characteristics in lambs finished under subtropical climatic conditions.

## 2. Material and Methods

All animal management procedures were conducted within the guidelines of federal locally approved techniques for animal use and care [11] and approved by the Ethics Committee of Faculty of Veterinary Medicine and Zootechnics from the Autonomous University of Sinaloa (Protocol #23012020).

### 2.1. Experimental Location

The experiment was conducted at the Universidad Autónoma de Sinaloa Feedlot Lamb Research Unit, located in Culiacán, México (24°46′13″ N and 107°21′14″ W). Culiacán is about 55 m above sea level and has a subtropical climate with a maximum temperature of 36 °C and a minimum temperature of 12 °C across the year.

### 2.2. Weather Measurement and Temperature Humidity Index (THI) Estimation

Climatic variables (ambient temperature and relative humidity) were obtained every hour from an on-site weather station (Thermo-hygrometer Avaly, Mod. DTH880, Mofeg S.A., Zapopan, Jalisco). The temperature humidity index (THI) was calculated using the following formula: THI = 0.81 × *T +* (RH/100) × (*T* − 14.40) + 46.40, where *T* is the temperature expressed in degrees Celsius and RH is the relative humidity [12].

### 2.3. Animals, Diets, and Sample Analyses

Forty Pelibuey × Katahdin crossbred intact male lambs (29.52 ± 4.79 kg initial live weight) were used in a 93 day growth performance experiment to evaluate the effects of treatments. Two weeks before initiation of the experiment, the lambs were subjected to an anthelmintic treatment (Closantel oral; 7 mg/kg, CLOSANTIL^®^ 5%, Laboratorio Chinoin, Mexico, City, Mexico), injected with 2 mL of vitamin A (500,000 IU, Synt-ADE^®^, Zoetis México, México City), and vaccinated against *Mannheimia haemolityca* (One Shot Ultra Zoetis México, México City). Upon initiation of the experiment, lambs were weighed before the morning meal (electronic scale; Torrey, Mod. EQM-400/800, TOR REY Electronics Inc, Houston TX, USA), and they were allotted to 20 pens according to their weight, where the pen was the experimental unit. Dietary treatments were randomly assigned to pens within blocks (five weight blocks) with two lambs per pen and five replicates per treatment. Pens were 6 m^2^ with overhead shade, automatic waterers, and 1 m fence-line feed bunks. Treatments consisted of a cracked corn-based basal total mixed finishing diet (Table 1) supplemented with eubiotics with a total dose of 3 g/lamb/day as follows:(1)no eubiotics (Control);(2)3 g of live *Saccharomyces cerevisiae*/lamb/day (2 × 10^10^ cfu/g; SC; Active Flora, ICC, São Paulo, Brazil);(3)3 g of mannan oligosaccharide plus b-glucans/lamb/day (MOS; 15% mannan oligosaccharide plus 25% b-glucans, *w*/*w*, Rumen Yeast, ICC, São Paulo, Brazil);(4)1.5 g/lamb/day SC plus 1.5 g/lamb/day MOS (SC+MOS).

The graphical description of experimental design are shown in Figure 1. The applied level of inclusion of probiotics and prebiotics was based on recommended feed additive label dosage (3 g/lamb/day), while the combination SC+MOS which was offered at 50% of each additive dose was based on the dose reported by Zapata et al. [7]. Lambs were weighed just prior to the morning feeding on days 1 and 93 of the experiment. Live weights (LW) on day 1 were converted to shrunk body weight (SBW) by multiplying LW by 0.96 to adjust for the gastrointestinal fill [13]. All lambs were fasted for 12 h before recording the final LW. Eubotics were hand-weighed using a precision balance (Ohaus, mod AS612, Pine Brook, NJ, USA), and were premixed for 5 min with minor ingredients (urea, limestone, and trace mineral salt) before incorporation into complete mixed diets. The final product was mixed with the remaining ingredients in a 1 m^3^ capacity horizontal mixer (Davis, H.C. Davis Sons, manufacturers, Bonner Spring, KS, USA). To avoid cross-contamination between treatments, the mixer was thoroughly cleaned between each batch. To ensure additive consumption, the total daily dosage per lamb was incorporated in 300 g of prepared diet provided in the morning feeding (all lambs were fed the basal Control diet in the afternoon feeding). Lambs were provided fresh feed twice daily at 8:00 a.m. and 2:00 p.m., in which the amount of feed provided in the morning feeding was constant (300 g/lamb), while feed delivered in the afternoon feeding was adjusted, allowing for a daily feed residual of refusal of ~50 g/kg. Residual feed was collected between 7:40 a.m. and 7:50 a.m. each morning and weighed. The adjustment to either an increase or a decrease in daily feed delivery was provided at the afternoon feeding. 

Feed samples were collected from batches of complete mixed diet. Feed refusals were collected daily and composited weekly for dry matter (DM) analysis (oven drying at 105 °C until no further weight loss; method 930.15) [14]. Feed samples were subjected to the following analyses: DM (oven drying at 105 °C until no further weight loss; method 930.15), CP (N × 6.25, method 984.13) by procedures described by AOAC [14], and NDF (corrected for NDF-ash, incorporating heat stable α-amylase; Ankom Technology, Macedon, NY) according to Van Soest et al. [15].

**Table 1 biology-10-01137-t001:** Composition of experimental diets offered ad libitum to the lambs.

	Treatments			
Item	Control	SC ^1^	MOS ^2^	SC+MOS ^3^
Ingredient composition (%)				
Sudan hay	10.00	10.00	10.00	10.00
Cracked corn	70.00	70.00	70.00	70.00
Soybean meal	9.50	9.50	9.50	9.50
Active Flora^®^	0	++	0	++
Rumen Yeast^®^	0	0	++	++
Molasses cane	5.00	5.00	5.00	5.00
Yellow grease	3.00	3.00	3.00	3.00
Protein–mineral premix ^4^	2.50	2.50	2.50	2.50
Chemical composition (%DM basis) ^5^				
Crude protein	13.43	13.43	13.43	13.43
Starch	52.91	52.91	52.91	52.91
Neutral detergent fiber	15.13	15.13	15.13	15.13
Ash	5.87	5.87	5.87	5.87
Gross energy, Mcal/kg	4.17	4.17	4.17	4.17
Calculated net energy (Mcal/kg) ^6^				
Maintenance	2.15	2.15	2.15	2.15
Gain	1.49	1.49	1.49	1.49

^1^ SC = live *Saccharomyces cerevisiae* at dose of 3 g/lamb/day (LSC; ActiveFlora, ICC, São Paulo, Brazil). ^2^ MOS = mannan oligosaccharide, b-glucans, and yeast metabolites at dose of 3 g/lamb/day (MOS, B-Glucans, and yeast metabolites; RumenYeast, ICC, São Paulo, Brazil). ^3^ SC+MOS = supplemented with 1.5 g/lamb/day LSC plus 1.5 g/lamb/day MOS. ^4^ Protein–mineral premix contained the following: crude protein, 50% urea-based; calcium, 28%; phosphorus, 0.55%; magnesium, 0.58%; potassium, 0.65%; NaCl, 15%; vitamin A, 1100 IU/kg; vitamin E, 11 UI/kg. ^5^ Dietary composition was determined by analyzing subsamples collected and composited throughout the experiment. Accuracy was ensured by adequate replication with acceptance of mean values that were within 5% of each other. ^6^ Calculated from tabular net energy (NE) values for individual feed ingredients [16].

### 2.4. Calculations

Average daily gain (ADG) was computed by subtracting the initial SBW (96% of full weight) [13] from the final SBW and dividing the result by the corresponding number of days on feed. Feed efficiency (weight gain-to-feed intake ratio, G:F) was computed by dividing ADG by the daily dry matter intake (DMI).

One approach for evaluation of the efficiency of dietary net energy (NE) utilization in growth performance trials is the observed-to-expected dietary NE ratio and the observed-to-expected DMI ratio [17]. On the basis of the measures of growth performance (observed DMI, ADG, and average SBW), the observed dietary net energy was calculated for each treatment by means of the following quadratic formula according to the procedure from Zinn et al. [17]:$$x=\frac{-b\pm \sqrt{b^{2}-4ac}}{2c},$$
where *x* is the observed NE_m_ (Mcal/kg), *a* = −0.41 EM, *b* = 0.877 EM + 0.41 DMI + EG, and *c* = −0.877 DMI.

EM is the energy required for maintenance (Mcal/day), and EG is the energy required for gain (Mcal/day), which were estimated using the following equations published by the NRC [16] as follows: EM = 0.056 × SBW^0.75^ and EG (energy gain, Mcal/day) = 0.276 × ADG × SBW^0.75^, in which the coefficient (0.276) was taken from NRC [18] assuming a mature weight of 113 kg for Pelibuey × Katahdin male lambs [19]. DMI corresponded to the average daily DMI (kg) registered for during the experiment.

According to the expected diet NE concentration calculated using the ingredient composition [16] and measures of growth performance, there was an expected energy intake. This estimation of expected DMI was performed on the basis of the observed average ADG and SBW, as well as on the NE values of the diet exposed in Table 1 according to the following equation: expected DMI, kg/day = (EM/NE_m_) + (EG/NE_g_), where EM is the energy required for maintenance (Mcal/day), EG is the energy required for gain (Mcal/day), and the NE_m_ and NE_g_ divisors are the corresponding NE values based on the ingredient composition of the experimental diet (Table 1).

### 2.5. Carcass Characteristics, Whole Cuts, and Shoulder Tissue Composition

All lambs were harvested on the same day. Lambs were stunned (captive bolt), exsanguinated, and skinned. The gastrointestinal organs were removed and weighed, the omental and mesenteric fat was weighed, and hot carcass weight (HCW) was recorded. After carcasses (with kidneys and internal fat included) were chilled at −2 to 1 °C for 24 h, the following measurements were obtained: (1) cold carcass weight (CCW); (2) body wall thickness (distance between the 12th and 13th ribs beyond the ribeye, five inches from the midline of the carcass); (3) measurement of subcutaneous fat (fat thickness) implemented over the 12th to 13th thoracic vertebrae; (4) longissimus muscle (LM) surface area, using a grid reading of the cross-sectional area of the longissimus muscle between 12th and 13th rib; (5) kidney, pelvic, and heart fat (KPH) removed manually and then weighed and reported as a percentage of the cold carcass weight [20]. Carcasses were split into two halves. The left side was fabricated into wholesale cuts, without trimming, according to the North American Meat Processors Association guidelines [21]. Rack, breast, shoulder, and foreshank were obtained from the foresaddle, and the loins, flank, and leg were obtained from the hindsaddle. The weight of each cut was subsequently recorded. The shoulder tissue composition was assessed using physical dissection according to the procedure described by Luaces et al. [22]. 

### 2.6. Visceral Mass Data

Components of the gastrointestinal tract (GIT), including tongue, esophagus, stomach (rumen, reticulum, omasum, and abomasum), pancreas, liver, gall bladder, small intestine (duodenum, jejunum, and ileum), and large intestine (caecum, colon, and rectum) were removed and weighed. The full GIT tract was then washed, drained, and weighed to get empty weights. The difference between full and washed digesta-free GIT was subtracted from the SBW to determine empty body weight (EBW). All tissue weights are reported on a fresh tissue basis. Organ mass is expressed as grams of fresh tissue per kilogram of final EBW. Full visceral mass was calculated by the summation of all visceral components (stomach complex + small intestine + large intestine + liver + lungs + heart), including digesta. The stomach complex was calculated as the digesta-free sum of the weights of the rumen, reticulum, omasum, and abomasum.

### 2.7. Statistical Analyses

Growth performance (weight gain, feed intake, and gain efficiency), dietary energetics, carcass data, and visceral mass data were analyzed as a randomized complete block design using the MIXED procedure of SAS software [23], where initial weight was the blocking criterion (blocks = 5), and pen was the experimental unit. The essence of this design is that the experimental units can be meaningfully grouped, with the number of units in a group or block being equal to the number of treatments. The object of grouping is to have the units in a block as uniform as possible so that observed differences will be largely due to treatments. Dietary treatments were randomly assigned to pens within blocks with two lambs per pen and five replicas per treatment according to the following statistical model:Yij = µ + B_i_ + T_j_ + ε_ij_,
where µ is the common experimental effect, B_i_ represents the initial weight block effect, T_j_ represents the dietary treatment effect, and ε_ij_ represents the residual error.

All the data were tested for normality using the Shapiro–Wilk test. Hot carcass weight was used as a covariate in evaluation of treatment effects on carcass characteristics and in the analysis of shoulder tissue composition. Treatment means were separated using the “honestly significant difference test” (Tukey’s HSD test). Treatment effects were considered significant at *p* ≤ 0.05 and were identified as trends at 0.05 < *p* ≤ 0.10.

## 3. Results

The temperature and relative humidity during the experiment are presented in Table 2. The average minimum and maximum estimated THI [11] was 70.49 and 86.72, respectively (Table 2). Daily maximal THI exceeded 80, which is considered as the “danger” or “emergency” range [24], for a few hours during each day of the 93 day study. During the first 5 weeks of the experiment, the daily THI averaged 76.15. From week 6 through the end of the study, daily THI averaged 80.13. The overall daily THI averaged 78.60, corresponding to “alert” conditions [24].

Treatment effects on growth performance and dietary energetic are shown in Table 3. There were no significant effects (*p* > 0.05) on DMI, averaging 1.17 ± 0.10 kg. Differences in ADG were not different among control, SC, and MOS treatments. However, DMI tended to be lower (*p* = 0.09) for SC and MOS treatments than for control. Consequently, G:F was greater (*p* = 0.01) for SC and MOS vs. the control. The DMI for SC+MOS was not different (*p* = 0.94) from that of the control group, but tended to be greater (*p* = 0.08) than that of SC or MOS sole supplementation. Lambs supplemented with SC+MOS had greater ADG (*p* = 0.04) and G:F (*p* < 0.01) than the control group. Compared to controls, supplementation with SC or MOS increased (*p* < 0.01) observed dietary net energy by 4.6% and 5.9%, respectively, while with the combination SC+MOS increased the observed dietary net energy by 7.2%. The combination SC+MOS improved ADG (*p* = 0.04), G:F (*p* = 0.02), and dietary energy (*p* = 0.04) compared to SC supplementation. When compared to MOS, the combination enhanced ADG (0.269 vs. 0.241 g/day, *p* = 0.04). This effect was mainly due to a tendency for increased DMI (1.213 vs. 1.121 kg/day, *p* = 0.06), as neither G:F nor dietary energy was appreciably affected (*p* > 0.12).

Treatment effects on carcass characteristics, shoulder tissue composition, whole cuts, and visceral mass are shown in Table 4, Table 5 and Table 6. Compared with the control and SC, supplementation with MOS and with SC+MOS increased (*p* ≤ 0.02) KPH (as a percentage of CCW). Compared with the control and SC, the combination SC+MOS increased (*p* = 0.02) visceral fat (g/kg EBW). On the other hand, SC supplementation tended to reduce (4.1%; *p* = 0.08) the relative intestinal mass compared to controls. Treatment effects on other carcass traits, shoulder tissue composition, and whole cuts were not significant.

Additionally, recent studies have reported metabolic changes that may enhance energy efficiency in stressed calves, including reduced plasma cortisol, NEFA, and urea-N concentration, and increased plasma glucose levels in stressed calves [8]. Reduced cellular oxidative stress has been reported for individuals receiving probiotic–prebiotic combination [8,10]. Under subtropical conditions, enhanced ADG of lambs supplemented with a probiotic–prebiotic combination was associated with increased plasma glucose (13%) and IGF-1 (35%) compared with lambs supplemented solely with either a probiotic or prebiotic [25].

## 4. Discussion

The THI ranges are presented in reference to *Bos taurus* cattle [24]. There are no specific codes for lambs; however, in wool lambs, the cattle THI codes may be indicative of potentially stressful ambient conditions [17,26,27]. Hairy lambs cope better with high ambient heat loads [28], as THI values climb to above 78 lamb growth performance and energetic efficiency is compromised [29,30]. In the present study, lambs were exposed to a daily average THI of 78 or greater for 61% of the experiment (8/13 weeks).

Decreased DMI is a notable response to elevated high ambient temperatures. In the present study, observed DMI was 5.3% less than predicted according to the intake model for feedlot lambs under thermal neutral conditions [13,17]. However, the expected reduction in DMI for feedlot lambs subjected to an ambient temperature of 30 °C is 12% [31]. The DMI of hairy lambs may be less impacted by ambient heat load [32] with no differences in DMI observed for Dorper × Kathahdin lambs during winter or summer months in a semiarid environmental [32,33]. However, Macías-Cruz et al. [29] observed a moderate DMI reduction (8.3%) when comparing Dorper × Pelibuey lamb growth performance for spring (THI = 68.12) vs. summer (THI = 81.80) months in a semiarid environment.

Responses to supplemental eubiotics in feedlot lambs have been inconsistent. In some cases, supplemental probiotics or prebiotics enhanced DMI and, in turn ADG, whereas, in other studies, G:F was enhanced without an effect on DMI [34,35]. The basis for these differences in treatment effects on DMI and weight gain may be associated with the climatic conditions, composition of diets fed, type of eubiotic, and/or levels of supplementation [5]. In the present experiment, the average intake of eubiotics was equivalent to 0.07 g of eubiotics/kg LW. This dosage level is within the previously observed effective range for positive growth performance responses to supplemental eubiotics [36,37]. Enhancement of both ADG and G:F without an effect on DMI is represented by a consistent growth performance response in feedlot lambs supplemented with a combination of probiotics with prebiotics [38,39,40]. Several arguments have been put forward to try to explain the greater benefit obtained with the combination. These hypotheses include stabilization of rumen environment, inhibition of pathogenic bacteria along the gastrointestinal tract, modulation of immune response, increase in fiber digestion, and enhanced nutrient uptake [36,37,41].

Reduced cellular oxidative stress and NEFA, and increased plasma glucose and IGF-1 are metabolic signals of greater energetic efficiency [42,43]. This is particularly relevant in growing–finishing animals exposed to environmental stressors (i.e., high ambient temperatures). Heat load has been associated with a 7% to 25% increase in maintenance energy requirements of lambs [29], mostly due to the metabolic adjustments for heat load dissipation. In healthy animals grown under non-stressful ambient conditions, the expected ratio of observed-to-expected dietary NE would be 1.0. That is, lamb ADG is consistent with DMI and energy density of the diet. If the ratio is greater than 1, the observed dietary NE is greater than anticipated according to the diet composition NRC [16], and efficiency of energy utilization is enhanced. In contrast, if the ratio is less than 1, energetic efficiency is less than expected. Therefore, the estimation of dietary energy intake and the ratio of observed-to-expected DMI reveal differences in efficiency of energy utilization independently of ADG. Accordingly, lambs that received either SC or MOS treatments utilized dietary net energy as expected, while utilization of dietary net energy by non-supplemented lambs was less than expected. Utilization of dietary net energy by lambs supplemented with the combination SC+MOS was also in line with the expectations. However, their greater DMI may be indicative of enhanced tolerance to conditions of high ambient heat load.

In animals under stress conditions (as in the present experiment), the energy requirements of maintenance may increase [30,44]. As a function of the efficiency of the partial utilization of energy for maintenance and gain, changes in maintenance requeriments can be estimated as follows: MQ = (NE_m_ × [DMI − {EG/NE_g_}])/SBW^0.75^, where NE_m_ corresponds to the NE values of the diet (Table 1) according to NRC [16] tables, and EG is the energy requirement for gain. Accordingly, in non-supplemented lambs, elevated THI increased the maintenance coefficient by 16% above 0.056 Mcal/SBW^0.75^ specified by the standard [18]. This increase is within the expected range of 7% to 25% greater maintenance requirement for heat-stressed cattle [30]. Applying the same equation, the estimated maintenance requirement due to SC and MOS treatments decreased in the same magnitude of 2.6% (0.0545 vs. 0.056). Supplementation with SC+MOS decreased the estimated maintenance requirement by 8.8% (0.049 vs. 0.056). These enhancements are consistent with previously observed positive effects of probiotic plus prebiotic combination supplementation on digestion and fermentation [7,39], as well as improved energy balance in cattle supplemented with eubiotics and their combination under conditions of stress [9,10]. From the perspective of efficiency dietary energy utilization for production, the use of eubiotics may be an additional strategy to reduce the negative effects of the high environmental heat load on the productivity of growing–finishing lamb. Under the conditions in which the experiment was carried out, the use of MOS proved to be more effective than SC. However, the combination SC+MOS brought about a greater enhancement in weight gain.

Consistent with previous studies, effects of probiotic or prebiotic supplementation on carcass characteristics [45,46] or wholesale cuts [45,47] of feedlot lambs were small and non-appreciable [45,48]. However, both supplemental MOS and SC+MOS affected fat depots, increasing KPH and visceral fat. The basis for these effects is not clear. In previous studies [46,47], MOS and SC+MOS tended to increase ruminal acetate-to-propionate ratio. Proportionally greater acetate may contribute to increased visceral fat deposition in ruminants [49]. Increased internal fat may also reflect the greater energy retention observed for the lambs receiving eubiotics.

Consistent with Belewu and Jimoh [50] and Raghebian et al. [51], there were no treatment effects on stomach complex, liver, heart, kidney, and lung mass. It has been demonstrated that probiotics inhibited the proinflammatory factors and triggered protective proteins in the intestinal cells, reducing inflammation and decreasing intestinal wall thickness in mammals [37,52]. Changes in intestinal wall thickness due to SC+MOS were inversely related to broiler health [53]. However, there is little information regarding effects of SC, MOS, or their combination on intestinal mass in ruminants fed with high-energy diets. García-Díaz et al. [54] reported that the combination of SC+MOS decreased the plasma concentrations of inflammatory factors in steers fed high-grain diets, speculating that this effect could contribute to a reduction in the inflammatory process in the rumen caused by consumption of the grain-based diets. In the present study, SC supplementation tended (*p* = 0.08) to reduce the relative intestinal mass, but MOS and combination SC+MOS did not. Teng and Kim [55] noted that further studies need to be conducted to elucidate the mechanisms of action of probiotics and prebiotics on the gut epithelial integrity and the immune system. The relative reduction in intestinal mass observed in the present study may be evidence of decreased inflammation of the intestinal wall with SC supplementation [37].

## 5. Conclusions

Eubiotics supplementation in finishing lambs under subtropical climatic conditions may assist in reducing the negative effects of high ambient heat load on the dietary energy utilization. Compared to controls, lambs receiving probiotics and/or prebiotics had greater gain efficiency and ratio of observed-to-expected diet net energy, with minimal effects on carcass characteristics, whole cuts, and visceral mass. Under the conditions in which this experiment was carried out, supplemental prebiotics (MOS) proved to be more effective than probiotics (SC), but the combination SC+MOS brought about a greater response in live weight gain. Consequently, the combination of probiotics (SC) plus prebiotics (MOS) appears to reinforce the positive effects of eubiotics.

## Figures and Tables

**Figure 1 biology-10-01137-f001:**
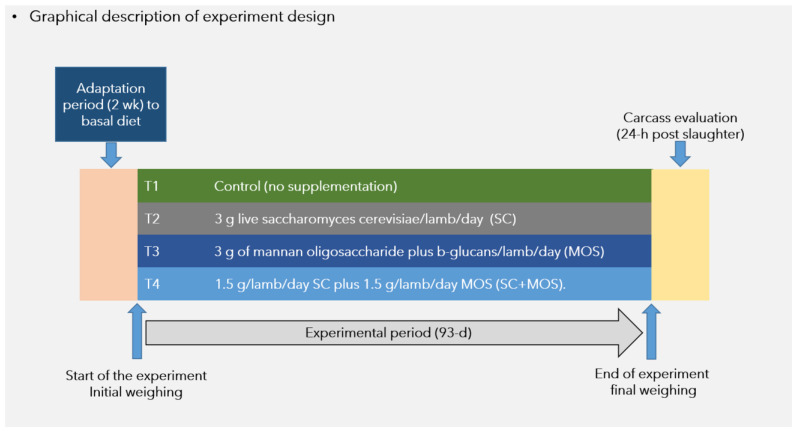
Description of assigned treatments and times for weight registration during experimental period and carcass evaluation.

**Table 2 biology-10-01137-t002:** Ambient temperature (Ta), relative humidity (RH), and calculated temperature humidity index (THI) ^1^ registered every hour and expressed as a weekly average.

Week	Mean T_a_ (°C)	Min T_a_ (°C)	Max T_a_ (°C)	Mean RH (%)	Min RH (%)	Max RH (%)	Mean THI ^1^	Min THI	Max THI
1	29.09 ± 0.9	24.56 ± 1.4	33.62 ± 0.5	39.70 ± 1.2	25.18 ± 0.3	54.21 ± 2.7	76.45 ± 0.8	66.86 ± 1.5	84.05 ± 1.5
2	28.01 ± 1.0	22.91 ± 1.3	33.10 ± 0.8	40.04 ± 1.3	24.89 ± 0.8	55.18 ± 2.4	75.30 ± 0.9	67.08 ± 1.4	83.53 ± 0.9
3	28.46 ± 1.1	22.67 ± 0.9	34.24 ± 0.3	37.09 ± 1.7	26.50 ± 0.2	47.68 ± 2.6	75.27 ± 1.0	66.96 ± 0.9	83.59 ± 0.6
4	27.98 ± 0.9	22.84 ± 0.7	33.12 ± 0.3	44.14 ± 2.6	27.14 ± 0.8	61.14 ± 2.7	75.93 ± 0.8	67.19 ± 0.8	84.67 ± 0.5
5	29.76 ± 1.0	23.23 ± 0.9	36.29 ± 0.3	40.83 ± 1.7	24.86 ± 0.3	56.79 ± 3.5	77.89 ± 0.9	67.41 ± 1.0	88.23 ± 0.7
6	30.36 ± 0.9	24.49 ± 0.9	36.24 ± 0.4	39.59 ± 1.8	25.61 ± 0.8	53.57 ± 3.6	78.14 ± 0.8	68.82 ± 1.0	84.98 ± 0.6
7	31.14 ± 0.9	25.44 ± 1.0	36.84 ± 0.4	38.73 ± 1.5	26.18 ± 0.6	51.28 ± 2.4	78.82 ± 0.8	69.90 ± 1.0	87.45 ± 0.5
8	31.89 ± 0.7	27.64 ± 0.9	36.14 ± 0.6	38.36 ± 1.7	25.14 ± 0.6	51.57 ± 3.7	79.50 ± 0.5	72.12 ± 0.9	87.75 ± 0.7
9	31.82 ± 0.8	26.66 ± 0.9	36.97 ± 0.3	38.45 ± 2.0	25.29 ± 0.6	51.61 ± 2.9	79.55 ± 0.7	71.10 ± 0.9	86.88 ± 0.5
10	32.03 ± 1.7	26.61 ± 0.9	36.46 ± 0.4	37.43 ± 1.7	26.07 ± 0.9	48.79 ± 3.4	79.56 ± 1.8	71.14 ± 1.0	87.99 ± 0.5
11	33.31 ± 0.7	29.45 ± 0.7	37.18 ± 0.3	41.29 ± 1.7	27.68 ± 0.9	54.89 ± 2.3	81.72 ± 0.5	74.42 ± 0.8	89.02 ± 0.4
12	33.60 ± 0.6	30.29 ± 0.5	36.90 ± 0.3	41.09 ± 1.9	28.75 ± 0.5	53.43 ± 2.0	81.91 ± 0.5	75.51 ± 0.6	88.31 ± 0.2
13	32.76 ± 0.8	29.93 ± 0.6	35.58 ± 0.9	46.50 ± 2.4	33.32 ± 2.2	59.68 ± 3.7	81.84 ± 0.6	75.82 ± 0.6	87.86 ± 1.1
Mean	30.79 ± 0.5	25.90 ± 0.9	35.67 ± 0.4	40.25 ± 1.1	26.66 ± 0.8	53.83 ± 1.8	78.60 ± 0.7	70.49 ± 1.0	86.72 ± 0.7

^1^ THI code (normal THI <74; alert 75 to 79; danger 79 to 84; emergency >84).

**Table 3 biology-10-01137-t003:** Effect of treatments on growth performance and dietary energy utilization of finishing lambs supplemented for 93 days with eubiotics alone or combined.

	Treatments ^1^							*p*-Value			
Parameter	Control	SC	MOS	SC+MOS	SEM	1 vs. 2	1 vs. 3	1 vs. 4	2 vs. 3	2 vs. 4	3 vs. 4
Days on test	93	93	93	93							
Pen replicates	5	5	5	5							
Live weight, kg/day^2^											
Initial	29.50	29.52	29.48	29.57	0.13	0.91	0.91	0.70	0.82	0.78	0.62
Final	51.99	51.79	51.86	54.56	0.80	0.86	0.90	0.04	0.95	0.03	0.03
Average daily gain, kg/day	0.242	0.241	0.241	0.269	0.008	0.84	0.93	0.04	0.91	0.03	0.04
Dry matter intake, kg/day	1.210	1.130	1.121	1.213	0.031	0.09	0.06	0.94	0.84	0.08	0.06
Feed efficiency (G:F), kg/kg	0.201	0.213	0.216	0.222	0.004	0.01	0.01	<0.01	0.36	0.02	0.12
Observed dietary net energy, Mcal/kg											
Maintenance	2.06	2.16	2.19	2.22	0.024	<0.01	<0.01	<0.01	0.33	0.04	0.22
Gain	1.40	1.48	1.51	1.54	0.021	<0.01	<0.01	<0.01	0.33	0.04	0.22
Observed to expected diet NE ^2^											
Maintenance	0.958	1.005	1.018	1.032	0.012	<0.01	<0.01	<0.01	0.33	0.04	0.22
Gain	0.940	0.993	1.013	1.034	0.011	<0.01	<0.01	<0.01	0.33	0.04	0.22
Observed to expected DMI	1.054	0.999	0.984	0.968	0.009	<0.01	<0.01	<0.01	0.33	0.04	0.22

^1^ SC = live *Saccharomyces cerevisiae* at dose of 3 g/lamb/day (LSC; ActiveFlora, ICC, São Paulo, Brazil); MOS = mannan oligosaccharide, b-glucans, and yeast metabolites at dose of 3 g/lamb/day (MOS, B-Glucans, and yeast metabolites; RumenYeast, ICC, São Paulo, Brazil); SC+MOS = supplemented with 1.5 g/lamb/day LSC plus 1.5 g/lamb/day MOS. ^2^ Initial and final shrunk weight is the full live weight reduced by 4% to adjust for gastrointestinal fill.

**Table 4 biology-10-01137-t004:** Effect of treatments on carcass characteristics of finishing lambs supplemented for 93 days with eubiotics alone or combined.

	Treatments ^1^							*p*-Value			
Parameter	Control	SC	MOS	SC+MOS	SEM	1 vs. 2	1 vs. 3	1 vs. 4	2 vs. 3	2 vs. 4	3 vs. 4
Hot carcass weight, kg	30.37	30.16	30.07	31.63	0.65	0.83	0.75	0.20	0.92	0.14	0.12
Dressing percentage	58.42	58.24	58.35	57.99	0.45	0.70	0.89	0.36	0.81	0.58	0.43
Cold carcass weight, kg	29.98	29.73	29.68	31.22	0.64	0.78	0.74	0.20	0.93	0.15	0.11
LM area, cm^2^	21.17	20.72	20.96	21.46	0.51	0.58	0.82	0.63	0.68	0.24	0.51
Fat thickness ^2^, cm	0.294	0.286	0.299	0.287	0.13	0.67	0.78	0.74	0.49	0.93	0.54
Kidney pelvic and heart fat, %	2.88	2.87	3.17	3.11	0.07	0.85	0.02	0.04	<0.01	0.03	0.52
Shoulder composition, %											
Muscle	63.28	63.20	62.60	63.24	0.67	0.98	0.53	0.97	0.54	0.98	0.55
Fat	18.50	18.88	18.26	18.78	0.60	0.71	0.78	0.75	0.53	0.96	0.55
Muscle-to-fat ratio	3.45	3.37	3.50	3.38	0.15	0.69	0.83	0.73	0.54	0.95	0.58

^1^ SC = live *Saccharomyces cerevisiae* at dose of 3 g/lamb/day (LSC; ActiveFlora, ICC, São Paulo, Brazil); MOS = mannan oligosaccharide, b-glucans, and yeast metabolites at dose of 3 g/lamb/day (MOS, B-Glucans, and yeast metabolites; RumenYeast, ICC, São Paulo, Brazil); SC+MOS = supplemented with 1.5 g/lamb/day LSC plus 1.5 g/lamb/day MOS. ^2^ Fat thickness over the center of the LM between of 12th and 13th ribs.

**Table 5 biology-10-01137-t005:** Effect of treatments on whole cuts of finishing lambs supplemented for 93 days with eubiotics alone or combined.

	Treatments ^1^							*p*-Value			
Whole cuts (as % of CCW)	Control	SC	MOS	SC+MOS	SEM	1 vs. 2	1 vs. 3	1 vs. 4	2 vs. 3	2 vs. 4	3 vs. 4
Neck	10.55	10.64	10.61	10.54	0.45	0.88	0.93	0.99	0.96	0.88	0.91
Shoulder IMPS207	15.61	15.45	15.28	15.36	0.20	0.58	0.25	0.39	0.54	0.75	0.76
Shoulder IMPS206	8.21	7.68	7.94	7.91	0.26	0.11	0.45	0.41	0.33	0.35	0.95
Leg IMPS233	25.98	25.43	24.96	25.97	0.50	0.39	0.16	0.99	0.46	0.39	0.17
Loin IMPS231	7.46	7.84	7.40	7.36	0.24	0.32	0.87	0.77	0.25	0.21	0.90
Rack IMPS204	7.73	7.49	7.63	7.69	0.25	0.52	0.92	0.90	0.59	0.60	0.99
Flank IMPS232	6.50	6.59	6.37	6.27	0.19	0.74	0.63	0.41	0.42	0.26	0.73
Breast IMPS209	3.76	3.62	3.81	4.04	0.33	0.29	0.87	0.36	0.21	0.11	0.45

^1^ SC = live *Saccharomyces cerevisiae* at dose of 3 g/lamb/day (LSC; ActiveFlora, ICC, São Paulo, Brazil); MOS = mannan oligosaccharide, b-glucans, and yeast metabolites at dose of 3 g/lamb/day (MOS, B-Glucans, and yeast metabolites; RumenYeast, ICC, São Paulo, Brazil); SC+MOS = supplemented with 1.5 g/lamb/day LSC plus 1.5 g/lamb/day MOS.

**Table 6 biology-10-01137-t006:** Effect of treatments on visceral mass of finishing lambs supplemented for 93 days with eubiotics alone or combined.

	Treatments ^1^							*p*-Value			
Parameter	Control	SC	MOS	SC+MOS	SEM	1 vs. 2	1 vs. 3	1 vs. 4	2 vs. 3	2 vs. 4	3 vs. 4
GIT fill, kg	4.33	4.40	4.41	4.22	0.40	0.92	0.89	0.85	0.98	0.76	0.74
EBW, % of full weight	92.00	91.90	92.86	92.72	0.67	0.92	0.82	0.55	0.90	0.49	0.42
Full viscera, kg	9.24	9.62	9.60	9.82	0.46						
Organs, g/kg of EBW											
Stomach complex	28.47	29.77	30.04	30.07	0.74	0.31	0.22	0.20	0.82	0.79	0.97
Intestines	41.91	40.17	40.31	40.40	0.59	0.08	0.12	0.15	0.53	0.48	0.85
Liver/spleen	15.09	14.94	15.34	15.41	0.44	0.67	0.25	0.21	0.14	0.11	0.92
Heart/lungs	21.26	20.21	20.95	21.91	0.71	0.36	0.82	0.61	0.49	0.16	0.47
Kidney	2.36	2.24	2.29	2.28	0.11	0.36	0.57	0.52	0.72	0.77	0.94
Visceral fat	37.34	37.37	39.52	40.81	1.38	0.86	0.22	0.02	0.16	0.02	0.19

^1^ SC = live *Saccharomyces cerevisiae* at dose of 3 g/lamb/day (LSC; ActiveFlora, ICC, São Paulo, Brazil); MOS = mannan oligosaccharide, b-glucans, and yeast metabolites at dose of 3 g/lamb/day (MOS, B-Glucans, and yeast metabolites; RumenYeast, ICC, São Paulo, Brazil); SC+MOS = supplemented with 1.5 g/lamb/day LSC plus 1.5 g/lamb/day MOS.

## Data Availability

Not applicable.

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
