# Peer review of "The Effects of Single or Combined Supplementation of Probiotics and Prebiotics on Growth Performance, Dietary Energetics, Carcass Traits, and Visceral Mass in Lambs Finished under Subtropical Climate Conditions"

_biology, 2021, doi:10.3390/biology10111137_

Round 1
Reviewer 1 Report
Manuscript biology-1377972, entitled “The effects of single or combined supplementation of probiotics and prebiotics on growth performance, dietary energetics, carcass traits, and visceral mass in lambs finished under subtropical climate conditions”
Recommendation: The above paper is not suitable for publication in its present form.
General comment
The article provides useful information about the effects of sole or combined supplementation of probiotics and prebiotics on growth performance, dietary energetics, carcass traits, and visceral mass in lambs finished under subtropical climate conditions. Although, the experiment is in general appropriately designed and implemented, there are some points that should be corrected or clarified.
My main concern is the presentation of results by the authors in some cases. For example, in L46-47, authors should be precise. Values for feed efficiency and dietary net energy are not the same in SC and MOS groups, so two percentages that show the enhancement should be provided and not one. Please also check L274-276, 251, 255-256, 364.
L33-34: “…to alleviate these adverse effects. In the present study probiotics or/and prebiotics supplementation improves dietary…”
L36: “reinforces” instead of “potentiate”
L47-48: “Compared to Controls, SC+MOS enhanced ADG (10%), G:F (9.5%) and dietary net energy (7.2%). Lambs fed SC+MOS had also greater…”
L50-51: “This effect could be attributed to the increased…”
L52: “significantly” instead of “appreciably”
L54: “…relative intestinal mass (as a proportion of empty body weight) when compared to controls.”
L55-57: Please rephrase (the same sentences as in simple summary)
L62-63: “The use of antibiotics as feed additives for growth promotion is increasingly restricted around the world [1]. Probiotics and prebiotics are promising alternatives…”
L67: “inhibit” instead of “inhibiting”
L72: “...were solely supplemented [5].”
L77: “a” instead of “at”
L78: “induce” instead of “enduce”
L79: “…reduced cortisol levels and cellular oxidative stress…”
L80: “…under stress conditions [7-9]. These…”
L111-113: Please delete “to evaluate the effects of probiotic, prebiotic and their combination on growth performance, dietary energetics and carcass characteristics”
L113: “subjected to an anthelmintic treatment” instead of “treated”
L119: “…and according to their weight were allotted to 20 pens…”
L121: “replicates” instead of “replicas”
L131: “The applied level of inclusion of probiotic and prebiotic was based…”
L141: “cross-contamination”
L204: “implemented” instead of “taken”
L205: Please delete “measure”
L241: Please delete “course of the”
L242: “The average minimum and…”
L243: “…exceeded 80 that is considered as “danger or “emergency” range…”
L249: “significant effects” instead of “treatment effect”
L250: “significant among” instead of “different for”
L252: “P=0.01”
L253-254: “…but tended to be greater than that of SC or MOS sole supplementation. Lambs…”
In Tables 3-6, in the row below treatments replace “None” with “Control” and “Item” with “Parameter”
L278: “…relative intestinal mass compared to controls. Treatment effects on other carcass traits, shoulder…”
L322: “These hypotheses include stabilization…”
L323: “modulation”
L326: “…including reduced…”
L327: “levels” instead of “in stressed calves”
L336: “solely” instead of “separately”
L345-347: “…diet composition by NRC [15] and efficiency of energy utilization is enhanced. In contrast, if ratio is less than 1, energetic efficiency is less than expected.”
L352: “…was also in line with the expectations.”
L402: “assist in reducing” instead of “help to reduce”
L403: What do you mean by “this source of eubiotics”? Please rephrase
Author Response
Response to REVIEWER 1.- Manuscript biology-1377972
AU: We are grateful to reviewers for the time and effort in helping improve the quality of the manuscript. The observations were wise and timely which permit the improvement substantially the manuscript. We have addressed the concerns in our revised manuscript accordingly.
All changes and correction made are highlighted in yellow in the corrected version of the manuscript.
RW: General comment
The article provides useful information about the effects of sole or combined supplementation of probiotics and prebiotics on growth performance, dietary energetics, carcass traits, and visceral mass in lambs finished under subtropical climate conditions. Although, the experiment is in general appropriately designed and implemented, there are some points that should be corrected or clarified.
RW: My main concern is the presentation of results by the authors in some cases. For example, in L46-47, authors should be precise. Values for feed efficiency and dietary net energy are not the same in SC and MOS groups, so two percentages that show the enhancement should be provided and not one. Please also check L274-276, 251, 255-256, 364.
AU: Following your suggestion, this observation was corrected in all the paragraphs indicated.
RW: L33-34: “…to alleviate these adverse effects. In the present study probiotics or/and prebiotics supplementation improves dietary…”
AU: Change was made as is suggested
RW: L36: “reinforces” instead of “potentiate”
AU: Change was made as is suggested
RW: L47-48: “Compared to Controls, SC+MOS enhanced ADG (10%), G:F (9.5%) and dietary net energy (7.2%). Lambs fed SC+MOS had also greater…”
AU: Sentence was rewording as is suggested
RW: L50-51: “This effect could be attributed to the increased…”
AU: Sentence was rewording as is suggested
RW: L52: “significantly” instead of “appreciably”
AU: Change was made as is suggested
RW: L54: “…relative intestinal mass (as a proportion of empty body weight) when compared to controls.”
AU: Sentence was rewording as is suggested
RW: L55-57: Please rephrase (the same sentences as in simple summary)
AU: The sentence was rephrased following your suggestion
RW: L62-63: “The use of antibiotics as feed additives for growth promotion is increasingly restricted around the world [1]. Probiotics and prebiotics are promising alternatives…”
AU: Sentence was rewording as is suggested
RW: L67: “inhibit” instead of “inhibiting”
AU: Done
RW: L72: “...were solely supplemented [5].”
AU: Correction was made
RW: L77: “a” instead of “at”
AU: Change was made as is suggested
RW: L78: “induce” instead of “enduce”
AU: Change was made as is suggested
RW: L79: “…reduced cortisol levels and cellular oxidative stress…”
AU: Sentence was rewording as is suggested
RW: L80: “…under stress conditions [7-9]. These…”
AU: Sentence was rewording as is suggested
RW: L111-113: Please delete “to evaluate the effects of probiotic, prebiotic and their combination on growth performance, dietary energetics and carcass characteristics”
AU: The phrase was deleted as suggested
RW: L113: “subjected to an anthelmintic treatment” instead of “treated”
AU: Sentence was rewording as is suggested
RW: L119: “…and according to their weight were allotted to 20 pens…”
AU: Sentence was rewording as is suggested
RW: L121: “replicates” instead of “replicas”
AU: Change was made as is suggested
RW: L131: “The applied level of inclusion of probiotic and prebiotic was based…”
AU: Sentence was rewording as is suggested
RW: L141: “cross-contamination”
AU: Done
RW: L204: “implemented” instead of “taken”
AU: Change was made as is suggested
RW: L205: Please delete “measure”
AU: Done
RW: L241: Please delete “course of the”
AU: Done
RW: L242: “The average minimum and…”
AU: “average” was inserted as suggested
RW: L243: “…exceeded 80 that is considered as “danger or “emergency” range…”
AU: The sentence was rephrased following your suggestion
RW: L249: “significant effects” instead of “treatment effect”
AU: Change was made as is suggested
RW: L250: “significant among” instead of “different for”
AU: Change was made as is suggested
RW: L252: “P=0.01”
AU: Thanks! Correction was made
RW: L253-254: “…but tended to be greater than that of SC or MOS sole supplementation. Lambs…”
AU: The sentence was rephrased following your suggestion
RW: In Tables 3-6, in the row below treatments replace “None” with “Control” and “Item” with “Parameter”
AU: Changes were made in all Tables following your suggestion.
RW: L278: “…relative intestinal mass compared to controls. Treatment effects on other carcass traits, shoulder…”
AU: Changes were made in all Tables following your suggestion
RW: L322: “These hypotheses include stabilization…”
AU: The sentence was rephrased following your suggestion
RW: L323: “modulation”
AU: Correction was made
RW: L326: “…including reduced…”
AU: Done
RW: L327: “levels” instead of “in stressed calves”
AU: Change was made as is suggested
RW: L336: “solely” instead of “separately”
AU: Change was made as is suggested
RW: L345-347: “…diet composition by NRC [15] and efficiency of energy utilization is enhanced. In contrast, if ratio is less than 1, energetic efficiency is less than expected.”
AU: The sentence was rephrased following your suggestion
RW: L352: “…was also in line with the expectations.”
AU: The sentence was rephrased following your suggestion
RW: L402: “assist in reducing” instead of “help to reduce”
AU: Change was made as is suggested
RW: L403: What do you mean by “this source of eubiotics”? Please rephrase
AU: Is regard of the specific type of probiotic and prebiotic used in the experiment. However, in order to be clearer, the sentence is rewording as follows in the corrected version of the manuscript: “Compared to Controls, lambs receiving probiotics and/or prebiotics had greater gain efficiency and ratio of observed-to-expected diet net energy

Reviewer 2 Report
I would like to suggest you to submit this article to more specify Journal like Journal of animal science or animal science Journal.
Author Response
Response to REVIEWER 2.- Manuscript biology-1377972
AU: We are grateful to reviewers for the time and effort in helping improve the quality of the manuscript. The observations were wise and timely which permit the improvement substantially the manuscript. We have addressed the concerns in our revised manuscript accordingly.
All changes and correction made are highlighted in yellow in the corrected version of the manuscript.
Comments and Suggestions for Authors
RW: I would like to suggest you to submit this article to more specify Journal like Journal of animal science or animal science Journal.
AU: First, we want to thank to you for the excellent comments given to our manuscript in your evaluation.
Regard to the journal, originally this manuscript was sent to the Animals Journal, but on recommendations from the Animals journal, we were directed to this prestigious journal. Given the high impact (and prestigious) of the Biology Journal, we are pleased to be in this process of possible publication.
Thank you for your wise comments.

Reviewer 3 Report
Dear Authors,
Many thanks for this valuable piece of work. I read it with interest and I found merit in it. However, some minor changes in my opinion are necessary.
I will not consider layout or English amends, which will be processed in further steps of article
editing, but only some suggestion which may increase the appealing of the paper for the reader.
The title is informative but the abstract should use more technical and reduced number of words (sounds too much discussing rather than sharp. For instance:
LL. 37-40: I would resume: The aim of this trial was to test the effects of the use of pro- and prebiotics alone or in combination in the diet of growing lambs. For this purpose...
L. 62: Please, change with 'The use of antibiotic growth promoters (AGPs).
L. 120: Please, change replicas with replicates
L. 130: Please, state if the 50%-50% is arbitrary in this case or derives from previous experiments.
On overall, the manuscript needs just minor revision. It is well displayed and structured.
Author Response
Response to REVIEWER 3.- Manuscript biology-1377972
AU: We are grateful to reviewers for the time and effort in helping improve the quality of the manuscript. The observations were wise and timely which permit the improvement substantially the manuscript. We have addressed the concerns in our revised manuscript accordingly.
All changes and correction made are highlighted in yellow in the corrected version of the manuscript.
Dear Authors, many thanks for this valuable piece of work. I read it with interest and I found merit in it. However, some minor changes in my opinion are necessary. I will not consider layout or English amends, which will be processed in further steps of article editing, but only some suggestion which may increase the appealing of the paper for the reader.
AU: Thanks for your wise comments
RW: The title is informative but the abstract should use more technical and reduced number of words (sounds too much discussing rather than sharp. For instance:LL. 37-40: I would resume: The aim of this trial was to test the effects of the use of pro- and prebiotics alone or in combination in the diet of growing lambs. For this purpose...
AU: Abstract was rewording in order to cover your suggestion
RW: L. 62: Please, change with 'The use of antibiotic growth promoters (AGPs).
AU: Change was made following your suggestion
RW: L. 120: Please, change replicas with replicates
AU: Correction was made
RW: L. 130: Please, state if the 50%-50% is arbitrary in this case or derives from previous experiments. AU: The combination SC + MOS which was offered at 50% of each additive dose was based on a dose previously reported. This clarification is already incorporated in the corrected version of the manuscript
RW: On overall, the manuscript needs just minor revision. It is well displayed and structured.
AU: Thank you for your positive feedback!

Round 2
Reviewer 1 Report
Authors made all the necessary amendments and I suggest the acceptance of the article.
Reviewer 2 Report
well revised, no more comments.